# Selective Induction of Intrinsic Apoptosis in Retinoblastoma Cells by Novel Cationic Antimicrobial Dodecapeptides

**DOI:** 10.3390/pharmaceutics14112507

**Published:** 2022-11-18

**Authors:** Vishnu Suresh Babu, Atish Kizhakeyil, Gagan Dudeja, Shyam S. Chaurasia, Veluchami Amutha Barathi, Stephane Heymans, Navin Kumar Verma, Rajamani Lakshminarayanan, Arkasubhra Ghosh

**Affiliations:** 1GROW Research Laboratory, Narayana Nethralaya Foundation, Bangalore 560099, India; 2Department of Cardiology, Cardiovascular Research Institute Maastricht (CARIM), Maastricht University, 6229 ER Maastricht, The Netherlands; 3Lee Kong Chian School of Medicine, Nanyang Technological University Singapore, Singapore 308232, Singapore; 4Retinoblastoma Service, Narayana Nethralaya, Bangalore 560099, India; 5Ophthalmology and Visual Sciences, Medical College of Wisconsin, Milwaukee, WI 53226, USA; 6Singapore Eye Research Institute, Singapore 169856, Singapore; 7Department of Pharmacy, National University of Singapore, Singapore 117559, Singapore

**Keywords:** DNA damage, antimicrobial peptides (CAPs), retinoblastoma

## Abstract

Host defense peptides represent an important component of innate immunity. In this work, we report the anticancer properties of a panel of hyper-charged wholly cationic antimicrobial dodecapeptides (CAPs) containing multiple canonical forms of lysine and arginine residues. These CAPs displayed excellent bactericidal activities against a broad range of pathogenic bacteria by dissipating the cytoplasmic membrane potential. Specifically, we identified two CAPs, named HC3 and HC5, that effectively killed a significant number of retinoblastoma (WERI-Rb1) cells (*p* ≤ 0.01). These two CAPs caused the shrinkage of WERI-Rb1 tumor spheroids (*p* ≤ 0.01), induced intrinsic apoptosis in WERI-Rb1 cells via activation of caspase 9 and caspase 3, cleaved the PARP protein, and triggered off the phosphorylation of p53 and γH2A.X. Combining HC3 or HC5 with the standard chemotherapeutic drug topotecan showed synergistic anti-cancer activities. Overall, these results suggest that HC3 and HC5 can be exploited as potential therapeutic agents in retinoblastoma as monotherapy or as adjunctive therapy to enhance the effectiveness of currently used treatment modalities.

## 1. Introduction

Nature-derived cationic antimicrobial peptides (CAPs) are the ancient armor against invading pathogens. CAPs are diverse in terms of amino acid composition, physicochemical properties as well as primary, secondary and tertiary structures. These host defense peptides are characterized by an overall positive net charge and a high abundance of hydrophobic amino acid residues. The bactericidal effect of CAPs is attributed to initial interactions between peptides and negatively charged bacterial cell walls, cell wall components, or cytoplasmic membranes [1]. CAPs have short amino acid residues with an amphipathic structure and a high abundance of arginine or lysine and hydrophobic residues [2]. The net positive charge of CAPs ensures their accumulation at an anionic microbial surface and high concentrations of hydrophobic residues facilitate their transit through the microbial outer membrane, eliciting selective antimicrobial activity while sparing the host cells [3]. CAPs, like polymyxin B [4], gramicidin S [5], and histatin [6] have been widely used in clinics due to their broad activity spectra against bacteria and fungi that are resistant to conventional medications [7,8]. 

Due to their cell surface charge selectivity, CAPs exhibit selective anticancer activity, and some of these do not cause unwanted bystander toxicity to healthy cells. CAPs that are positively charged with amphipathic features recognize cancer cells by the presence of negatively charged phosphatidylserine (PS) on their surface. They can cause membrane destabilization, disruption, and permeation. They can act on intracellular targets and induce apoptosis [9]. The selectivity of CAPs for cancerous cells over healthy cells is also ascribed to the differences in mammalian cell membrane composition, fluidity, and surface area of cells. For example, the LHH1 peptide with 16 amino-acid residues identified in *Lactobacillus casei* HZ1, induced apoptosis in the colon, gastric and nasopharyngeal cancer cells by disrupting their cellular membranes [10]. Synthetically modified cationic peptides, KKK and KGK displayed anticancer activity through caspase-mediated apoptosis in breast, prostate, and pancreatic cancer cells [11].

Retinoblastoma (Rb) is pediatric eye cancer, predisposed primarily by mutations in RB1 developing cone cells of the retina [12]. The main clinical goal in treating children with Rb involves preserving life and vision. Over the years, Rb management strategies have evolved from enucleation [13], plaque brachytherapy [14], external beam radiation [15], cryotherapy [16], and photocoagulation [17] to chemo reduction [18]. Targeted chemotherapy administered by intra-vitreal [19] or intra-ophthalmic artery [20] using melphalan, carboplatin, topotecan, vincristine, or their combination with focal therapy, achieved favorable tumor management and globe salvage in minimal or moderate Rb. However, these chemotherapeutic agents have a variable response in advanced Rb and cause undesirable toxicity to the developing retina [21]. Failure to control the recurrence of subretinal and vitreous seeds related to the main tumor is also a major drawback of targeted chemo-reduction strategy in advanced Rb [22]. Unlike targeted therapy, systemic co-administration of these agents is effective in managing early-stage Rb [23] and in preclinical models [24]. However, the blood-retinal barrier (BRB) hinders their penetration and diffusion to the tumor site. Small molecule drugs like Nutlin-3a [25], HDAC inhibitors (vorinostat, trichostatin A, entinostat) [26], BAY 61-3606 and R406 [27] have also been tested in Rb preclinical models, but their poor pharmacokinetic performance and systemic toxicity made them unattractive for Rb management. Therefore, adjuvants to enhance the local effects of current chemotherapeutic treatments and spare the healthy retina and globe from bystander toxicity is a critical requirement.

Here, we investigated a panel of de novo-designed CAPs for their antimicrobial and anticancer properties in the context of Rb and probed the mechanism of their tumoricidal functions as well as their synergism with topotecan.

## 2. Materials and Methods

### 2.1. Chemicals and Peptides 

Topotecan hydrochloride hydrate was purchased (cat#1672257, Sigma Aldrich, St. Louis, MO, USA) and was dissolved in 0.1% DMSO to make a final stock concentration of 20 mg/mL. Six different hyper-charged peptides (named in the current study as HC1 to HC6) were used. CAPs with 12 amino acid residues in length containing D- or L- lysine and arginine repeats were purchased from M/s. Peptide Mimotopes Pty Ltd., Victoria, Australia. All the peptides with a ≥95% purity were used as received. High-performance liquid chromatography (HPLC) of the purified peptides provided by the supplier is presented as Appendix A.

### 2.2. Antimicrobial Properties of CAPs

The minimum inhibitory concentrations (MIC) of cationic peptides were determined using the broth microdilution method in a 96-well microtiter plate (Cellstar, Greiner Bio-One, Austria). The inoculum was prepared by suspending overnight grown isolated colonies selected from a Tryptic Soy Agar (TSA) plate in water for injection. The concentration of the inoculum suspension was adjusted to match that of the 0.5 McFarland’s standard (10^8^ CFU/mL) and further diluted 150-fold by adding 50 μL of suspension into 7.5 mL of Mueller−Hinton broth to obtain an inoculum of 10^5^ to 10^6^ CFU/mL. Lyophilized peptide obtained from the manufacturer was dissolved in water and adjusted to a final concentration of 0.5 to 256 μg/mL in twofold serial dilutions. A total of 100 μL of the solution was then mixed with an equal volume of bacterial inoculum prepared before in 96-well plates. The plates were then incubated at 35 °C for 24 h, and the optical density at 600 nm (OD_600_) was determined using a TECAN microplate reader. Experiments were performed in duplicates and the MIC was taken to be the lowest peptide concentration exhibiting no visible microbial growth after incubation. 

For the membrane depolarization assay, suspensions of *P. aeruginosa* cells (cat#9027, ATCC, Manassas, VA, USA) grown overnight were harvested, resuspended in 5 mM HEPES buffer at pH 7.4, and adjusted to an OD600 of 0.2. The cytoplasmic membrane potential-sensitive probe DiSC3-5 (cat#D306, Sigma-Aldrich, St. Louis, MO, USA) was added to the inoculum to a final concentration of 10 μM, and the mixture was incubated at room temperature (RT) for 1 h. The suspension was transferred into a 10-mm stirring quartz cuvette, and 1 μL of each polymer in different concentrations (expressed as ×MIC) was added after a stable signal was detected. Fluorescence intensities at excitation and emission wavelengths of 622 nm and 670 nm, respectively, were monitored until a plateau was reached by using a Quanta Master spectrophotometer (Photon Technology International, Birmingham, NJ, USA) with slit widths of 0.5 nm. The fluorescence intensity values after the addition of Triton X (0.1%, *w*/*v*) were taken as maximum depolarization, and the percentage of depolarization was calculated.

### 2.3. Cell Culture

The Rb cell line WERI-Rb1 was obtained from the American Type Culture Collection (cat#HTB-169, ATCC, Manassas, VA, USA). Cells were maintained in Roswell Park Memorial Institute medium (cat#11875093, Gibco, Grand Island, NY, USA) supplemented with 10% (*v*/*v*) fetal bovine serum (cat#A4766801, Gibco, Grand Island, NY, USA) and 1% penicillin-streptomycin (5000 U/mL; cat#15070063, Gibco, Grand Island, NY, USA), as described [28]. The cells were cultured at 37 °C in a 90% humidified incubator with 5% CO_2_. Human adult retinal pigment epithelial cells (ARPE-19) (cat#CRL-2302, ATCC, Manassas, VA, USA) were cultured in Dulbecco’s Modified Eagle’s Medium/Nutrient Mixture F12 (cat#11710035, Gibco, Grand Island, NY, USA) supplemented with 10% fetal bovine serum (cat#A4766801, Gibco, Grand Island, NY, USA) and 1% Penicillin-Streptomycin (5000 U/mL; cat#15070063, Gibco, Grand Island, NY, USA).

### 2.4. Generation of Multi-Cellular Tumor Spheroids of WERI-Rb1 Cells

Spheroids were developed using the hanging drop protocol [29] on low attachment U-bottom 96-well plate. The single cell suspension of 5 × 10^2^ WERI-Rb1 cells in 100 µL medium was loaded in each well of a 96-well plate. The cells were cultured at 37 °C in a 90% humidified incubator with 5% CO_2_ for 7 days for the generation of tight and regular tumor spheroids. Seven-day-old spheroids (*n* = 9 per condition) were treated with 100 μL of freshly prepared medium containing an IC_50_ dose of topotecan (Appendix A), HC3, or HC5. The cultures were then incubated for 48 h prior to high content analysis (HCA) and immunofluorescence (IF) assays.

### 2.5. Cell Viability Assays

(a)
*Trypan blue viability assay:*


WERI-Rb1 cells (1 × 10^5^) were treated with increasing concentrations (7.7 µM, 36.25 µM, 72.5 µM, 145 µM, or 290 µM) of a panel of six HC peptides (HC1, HC2, HC3, HC4, HC5, and HC6). Nocodazole (10 µM) treatment was used as a positive control for the study, with untreated cells as the negative control. All treatments were performed at 37 °C in a 90% humidified incubator with 5% CO_2_ for 48 h. After 48 h post-treatment, cells were stained with 0.4% trypan blue (cat#15250061, Invitrogen, Waltham, MA, USA) and the number of live and dead cells for each time point was estimated using the hemocytometer cell counting protocol [30]. The percentage of cell viability was calculated by dividing the live cell count by the total cell count.

(b)
*MTS [3-(4,5-Dimethylthiazol-2-yl)-5-(3-carboxymethoxyphenyl)-2-(4-sulfophenyl)-2H-tetrazolium] cell viability assay:*


WERI-Rb1 and ARPE-19 cells were seeded at a concentration of 2 × 10^3^ cells/well in a 96-well plate and cultured at 37 °C in a humidified 5% CO_2_ incubator. After seeding the cells, they were treated with various concentrations of CAPs and further incubated for 48 h at 37 °C. Cell viability was assessed after 48 h using the Cell-Titer 96^®^ Aqueous One Solution Cell Proliferation Assay (cat#G3582, Promega, WI, USA) according to the manufacturer’s protocol and described earlier [31]. Briefly, 10 µL of MTS reagent was added to each well and the cells were incubated at 37 °C for 2 h before the experiment end time points. Percentage cell viability was calculated from the absorbance measured at 490 nm using a TECAN microplate reader.

### 2.6. Annexin V/PI

The FITC-Annexin V Apoptosis Detection Kit (cat#556547, BD Bioscience, Heidelberg, Germany) was used according to the manufacturer’s protocol. In brief, WERI-Rb1 cells treated with CAPs or topotecan were washed twice with cold phosphate-buffered saline (PBS) and resuspended in binding buffer at a final density of 10^6^ cells/mL. FITC-Annexin (5 μL) and propidium iodide (5 μL) were added to 100 μL of the cell suspension containing 10^5^ cells. The cell suspension was mixed by a gentle vortex and then incubated for 15 min at RT in the dark. Subsequently, 400 μL of binding buffer was added and cells were analyzed by flow cytometry using FACS Calibur (BD Bioscience, Heidelberg, Germany) and BD FACSDiva software (BD Bioscience, Heidelberg, Germany). 

### 2.7. High Content Imaging of Tumor Spheroids

WERI-Rb1 cells were stained with orange plasma membrane stain as per the manufacturer’s protocol (Cell Mask Orange Plasma membrane stain; cat#C10045, Invitrogen, Waltham, MA, USA) prior to the development of spheroids. The stained WERI-Rb1 cells were used for spheroid formation using the hanging drop protocol [29] on a low attachment U-bottom 96-well plate. Tight and regular spheroids (*n* = 9) were further exposed to IC_50_ doses of topotecan, HC3 (IC_50_ = 77.2 µL/mL), or HC5 (IC_50_ = 65.2 µL/mL) for 48 h. The images were acquired using the automated high-content imaging system, IN Cell Analyzer 6000 (GE Healthcare, Singapore), equipped with a laser-based confocal platform having a 2× objective. The automatically captured single well imaging of spheroids and a Z stack of 11 images were separated by 50 µm total 500 µm depth for spheroids, using maximum intensity projection (MIP) acquisition and single-channel filter at 567 nm. To reduce data storage and image analysis time, we analyzed 2D projection images of tumor spheroids for multiple treatment conditions. Spheroid area and shape were automatically quantified by HCA using the IN-Cell 6000 software (GE Healthcare, Singapore). The mean area (µm^2^) of each spheroid was calculated to elucidate the drug/peptide response. 

### 2.8. Peptide/Drug Synergy Testing

Drug interactions were assessed using a modified checkerboard microdilution method [32] and MTS assay. WERI-Rb1 cells were seeded at a concentration of 2×10^3^ cells/well in a 96-well plate and cultured at 37 °C in a humidified 5% CO_2_ incubator prior to the assay. IC_50_ doses of HC3, HC5, and topotecan were prepared and further diluted to various fractions of IC_50_ (IC_50_/2, IC_50_/4, IC_50_/8). To test the interaction of the drugs, each peptide was tested against topotecan with varying IC_50_ values for 48 h. The percentage of viable cells in each condition was determined using the MTS assay and the percentage of viability was represented as a heat map. The fractional inhibitory concentration (FIC) of the peptides and topotecan were calculated and interpreted as per the standard procedures [33]. The Fractional Inhibitory Concentration Index (FICi) was determined by using the following equation: $$\mathrm{FICi}=\frac{{\mathrm{IC}}_{50}\;\mathrm{combination}}{{\mathrm{IC}}_{50}\;\mathrm{Topotecan}}+\frac{{\mathrm{IC}}_{50}\;\mathrm{combination}}{{\mathrm{IC}}_{50}\;\mathrm{Peptide}}$$

FICi value of ≤0.5 was considered synergistic; a value of >0.5–1 indicated an additive effect of the two drugs, and a FICi value of >1 displayed the antagonism of the two drugs [34].

### 2.9. Western Blot Analysis

WERI-Rb1 cells were seeded in 6-well cell culture plates (1 × 10^6^ cells/well), incubated at 37 °C in a humidified 5% CO_2_ incubator, and were treated with IC_50_ doses of topotecan, HC3. or HC5. After 48 h, cells were lysed in RIPA buffer (20 mM Tris pH 8.0, 0.1% SDS, 150 mM NaCl, 0.08% sodium deoxycholate, 1% NP40 supplemented with one tablet of protease inhibitor and phosphatase inhibitor (Roche). 20 µg of total protein was loaded per lane and separated by SDS-PAGE. The separated proteins on the gel were transferred to PVDF membrane and were probed for antibodies against γH2A.X (1:1000, cat#2595, Cell Signaling Technology, Danvers, MA, USA), p-γH2A.X (Ser139) (1:1000, cat#2577, Cell Signaling Technology, Danvers, MA, USA), p-BCL2(Ser70) (1:1000, cat#2827, Cell Signaling Technology, Danvers, MA, USA), BCL2 (1:1000, cat#2872, Cell Signaling Technology, Danvers, MA, USA), Bax (1:1000, cat#2772, Cell Signaling Technology, Danvers, MA, USA), p-ATM(Ser1981) (1:1000, cat#5883, Cell Signaling Technology, Danvers, MA, USA), ATM (1:1000, cat#2873, Cell Signaling Technology, Danvers, MA, USA), p-Chk2(Thr68) (1:1000, cat#2661, Cell Signaling Technology, Danvers, MA, USA), Chk2 (1:000, cat#2662, Cell Signaling Technology, Danvers, MA, USA), p-p53(Ser15) (1:1000, cat#9284, Cell Signaling Technology, Danvers, MA, USA), p-p53(Ser46) (1:1000, cat#2521, Cell Signaling Technology, Danvers, MA, USA), p53 (1:1000, cat#2524, Cell Signaling Technology, Danvers, MA, USA), Cleaved caspase 9 (1:1000, cat#9505, Cell Signaling Technology, Danvers, MA, USA), Cleaved caspase 3 (1:1000, cat#9661, Cell Signaling Technology, Danvers, MA, USA), Cleaved PARP (1:1000, cat#5625, Cell Signaling Technology, Danvers, MA, USA), and GAPDH (1:2000, cat#ab9485, Abcam, Cambridge, UK). Proteins of interest were detected with HRP-conjugated sheep anti-mouse (1:5000, cat#7076, Cell Signaling Technology, Danvers, MA, USA) or anti-rabbit IgG antibody (1:5000, cat#7074, Cell Signaling Technology, Danvers, MA, USA) and visualized with the Clarity ECL Western blotting substrate (Bio-Rad, Hercules, CA, USA) according to the provided protocol.

### 2.10. Immunofluorescence Microscopy

WERI-Rb1 spheroids were treated with an IC_50_ dose of topotecan, HC3, or HC5 for 48 h. After the treatment time point, the spheroids were carefully rinsed with ice-cold PBS and fixed with 4% paraformaldehyde for 10 min at RT. The spheroids were permeabilized with 0.1% Triton X-100 for 10 min at RT. To assess DNA damage, spheroids were incubated with the p-γH2A.X(Ser139) antibody (1:200, cat#2577, Cell Signaling Technology, Danvers, MA, USA) diluted in 5% BSA/PBS for 1 h at RT. Spheroids were carefully washed twice with 0.02% Tween 20 and 1% BSA in PBS, followed by incubation with Alexa Fluor 488 conjugated anti-rabbit (1:500, cat#A-11008, Invitrogen, Waltham, MA, USA) and Alexa Fluor Phalloidin 647 which stains for F-actin, conjugated with anti-mouse (1:500, cat#A22287, Invitrogen, Waltham, MA, USA) for 45 min at RT. After three washes with 0.02% Tween 20 and 1% BSA in PBS, the spheroid nuclei were stained with Hoechst 33342 for 5 min at RT (1:5000, cat#H1399, Invitrogen, Waltham, MA, USA). The spheroid images were visualized and captured using an upright fluorescent microscope. 

### 2.11. Cytochrome C Release Assay

WERI-Rb1 cells were seeded in 6-well cell culture plates (1 × 10^6^ cells/well) incubated at 37 °C in a humidified 5% CO_2_ incubator and were treated with the IC_50_ dose of topotecan, HC3, or HC5 for 48 h. After the treatment time point, the cell pellets were washed twice with ice-cold PBS and the pellets were lysed with cytoplasmic extraction buffer followed by mitochondrial extraction buffer using the cytochrome c releasing apoptosis assay kit (cat#ab65311, Abcam, Cambridge, UK) as per the manufacturer’s protocol. 20 µg of mitochondrial and cytosol protein fractions were separated on 12% SDS PAGE and were transferred to a PVDF membrane. Both mitochondrial and cytosol fractions were probed for cytochrome c, using an antibody against cytochrome c (1:1000, Abcam) provided with the assay kit. Anti-VDAC1 (1:1000, cat#ab15895, Abcam, Cambridge, UK) was used as a mitochondrial fraction loading control and GAPDH (1:2000, cat#ab9485, Abcam, Cambridge, UK) was used as cytosol fraction loading control for the assay.

### 2.12. Statistical Analysis

Statistical analysis was performed using GraphPad Prism 8. Data are presented as mean ± s.d. unless indicated otherwise, and *p*< 0.05 was considered statistically significant. For all representative images, results were reproduced at least three times in independent experiments. For all quantitative data, the statistical test used is indicated in the legends. Briefly, normal distribution was first determined using the Shapiro–Wilk normality test. Data determined to be parametric were analyzed by ordinary one-way analysis of variance (ANOVA) with Tukey’s multiple comparisons test for select pairwise comparisons as indicated. Data determined to be nonparametric were analyzed by Kruskal–Wallis test with Dunn’s multiple comparisons for select pairwise comparisons as indicated. All tests were two-tailed unless otherwise indicated.

## 3. Results

### 3.1. Design and Antimicrobial Screening of Novel CAPs

An important sequence hallmark of antimicrobial peptides is the presence of the cationic amino acid residues arginine and lysine. These residues enhance the interactions of peptides with the negatively charged bacterial membrane [1,2]. The guanidine group present in arginine promotes the translocation of cell-penetrating peptides [3,4,5]. In our previous work, we compared the cell selectivity of various cationic polymers against microbial and mammalian cells [35]. We identified ε-polylysine as the polymer with high cell selectivity [35]. In another study, Shima et al. showed that a minimum of nine iso-lysine linkages are necessary to impart appreciable antimicrobial activity [36]. As the number of repeats increased to >12, a plateau was observed. These results suggest that a minimum of 12 cationic residues are necessary to obtain reasonably good antimicrobial activities based on MIC values (Appendix A). 

Natural antimicrobial peptides contain a high abundance of arginine and lysine contents, suggesting that the interaction of cationic residues with the negatively charged bacterial surface is important for antimicrobial activity [37]. To infer the role of cationic residues on antimicrobial activity, we designed a series of twelve residue peptides containing alternative α-/ε-lysine and arginine residues, labeled as hyper-charged (HC) peptides (Table 1). 

We next determined the MIC of HC peptides against a panel of Gram-negative (*P. aeruginosa*) and Gram-positive (*S. aureus/MRSA*) bacteria [38] (Appendix A). Mean (± SEM) values in the range of 20 to 35 µM were obtained for HC peptides against various *P. aeruginosa* strains (Figure 1a). One-way ANOVA by Dunnett’s multiple comparison test indicated that the mean MIC values of HC3, HC4 and HC5 were statistically significant compared to the HC1 peptide. Against *S. aureus/MRSA* strains, the mean (± SEM) values were in the range of 16 to 25 µM for HC peptides. No significant difference, however, was observed in the antimicrobial properties of peptides against *S. aureus/MRSA* strains (Figure 1b). 

The lethal action of CAPs has been attributed to the interaction of peptides with the negatively charged membrane components of the cytoplasmic membrane, leading to rapid dissipation of membrane potential and concomitant release of vital intracellular components. Therefore, we determined the ability of peptides to depolarize the cytoplasmic membrane of *P. aeruginosa* strains, using a membrane potential sensitive probe and estimated the degree of depolarization (Appendix A). The results suggested that peptides containing both D- or L- ε-lysyl residues elicited higher membrane potential loss than peptides containing α-L-lysine residues (Figure 1c). 

### 3.2. CAPs Display Anticancer Activities on Rb Cells 

The homopolymers of cationic amino acid residues exhibit varying degrees of cell penetration: poly arginine displays superior cell penetration compared to other homopolymers of similar length [39]. A minimum of six arginine residues are required for energy-dependent cellular uptake [39]. Since apoptosis is characterized by phosphatidyl serine externalization, a negatively charged lipid on the cell membrane, increased transmembrane potential, membrane fluidity, and surface area [40], we sought to investigate whether the HC peptides could cause such changes in WERI-Rb1 cells. We first investigated the cytotoxic effects of peptides by trypan blue and propidium iodide (PI) exclusion assays. Among the peptides tested, HC3 and HC5 displayed potent anticancer activity with increasing doses, causing 40% cell death (*p* < 0.001) at a low dose and 60% cell death at a high dose after 48 h treatments (Figure 2a). However, the other CAPs (HC1, HC2, HC4, HC6) induced moderate cytotoxic effects or required high doses (500 µM) to achieve 50% cell lethality. The IC_50_ values of HC peptides ranged between the two prolific anticancer drugs topotecan (IC_50_ = 10 nM, Appendix A) and doxorubicin (IC_50_ = 2.5 mM, Appendix A). Furthermore, HC3 and HC5 did not display any cytotoxic effect on non-malignant ARPE-19 cells (Figure 2c), highlighting the peptide’s ability to discriminate between cancerous and non-cancerous cells. To determine the percentage of apoptotic cell death in WERI-Rb1 cells upon CAPs treatment, we used the Annexin V-FITC/PI assay as described earlier [41]. We observed a significantly high percentage of apoptotic cell death in HC3- and HC5-treated WERI-Rb1 cells (Figure 2d).

### 3.3. CAPs Reduce Rb Tumor Spheroid Assembly and Growth

HC3 and HC5 treatment effectively shrunk and decreased the Rb spheroid size. Between the two peptides, HC3 was more effective in decreasing the spheroid size than HC5 (*p* < 0.001) at 48-h treatment (Figure 3a). Topotecan (used as a control) also disrupted Rb spheroids indicating its hypersensitivity to DNA damage-induced cell death (Figure 3a,b) [42].

### 3.4. CAPs Induce DNA Damage and Direct WERI-Rb1 Cells towards Apoptosis

Next, we investigated the effect of HC3 and HC5 in inducing DNA damage in WERI-Rb1 cells. We observed induction of γH2A.X phosphorylation at Ser139 in topotecan-treated cells while immunoblots of HC3 or HC5 treated cells did not show any activation of γH2A.X(Ser139) (Figure 4a). HC3 and HC5 induced phosphorylation of ATM (ataxia-telangiectasia mutated) at Ser1981 and CHK2 at Thr68 during 48-h treatment, indicating double-strand DNA breaks (Figure 4b). CAPs activated p53 by inducing phosphorylation at Ser15 as well as Ser46 residues (Figure 4b), which are indicative of DNA damage and initiation of an irreversible cell death program [43,44]. Moreover, we observed increased phosphorylation of γH2A.X in topotecan and CAPs-treated WERI-Rb1 spheroids (Figure 4c), further confirming a role of CAPs in activating cellular DNA damage response and apoptosis.

### 3.5. CAPs Selectively Disrupt the Pro-Apoptotic and Anti-Apoptotic Balance in WERI-Rb1 Cells

Topotecan induces cell apoptosis by altering the Bcl2/Bax activation/expression [45]. Treatment of WERI-Rb1 cells with CAPs showed reduced phosphorylation of anti-apoptotic Bcl2 protein at Ser70 and increased expression of pro-apoptotic protein Bax (Figure 5a). Bcl2 phosphorylation at Ser70 was stable in controls compared to its reduced expression in topotecan treatment conditions. CAPs caused the release of cytochrome c from the mitochondria to the cytoplasm, indicative of irreversible induction of cell death (Figure 5b). However, HC3 and HC5 did not trigger any apoptotic events in ARPE-19 cells, while topotecan induced Bax expression in the mitochondrial fraction of ARPE-19 (Figure 5c) and cytochrome c release from mitochondria to the cytoplasm (Figure 5c). HC3 and HC5 were able to enhance Bax expression and disrupt cell survival balance selectively in Rb cells, highlighting the excellent cancerous cell selectivity of the peptides.

### 3.6. CAPs Localize to Mitochondria of WERI-Rb1 Cells and Initiate Programmed Cell Death

CAPs cause mitochondrial damage by disturbing the balance of Bcl2/Bax expression and the cytochrome c release from the inner mitochondrial membrane to the cytoplasm. To understand the CAPs localization in the mitochondria, we used fluorescein amidites (FAM) labeled HC3 in WERI-Rb1 cells (Figure 6a) and ARPE-19 cells (Figure 6b). FAM-HC3 treated WERI-Rb1 cells displayed localization of HC3 along with VDAC1 in the mitochondrial outer membrane (Figure 6a) and caused cytotoxic effects and apoptotic events, evident from Figure 2, Figure 3, Figure 4 and Figure 5. Mitochondrial localization of FAM-HC3 in ARPE-19 cells did not show any cytotoxicity (Figure 6b).

### 3.7. CAPs Drive Apoptosis through the Intrinsic Pathway 

The release of cytochrome c in the cytoplasm forms apoptotic initiator complexes that activate the caspase cascade [46]. Upon CAPs treatment in WERI-Rb1 cells, we observed the activation of caspase 9 with the presence of a cleaved protein sizes of 37 kDa and 35 kDa (Figure 7a). No cleaved bands were observed in the control, while topotecan treatment conditions showed cleaved caspase 9 (Figure 7a). CAPs also cleaved caspase 3 with protein sizes of 19 kDa and 17 kDa in the immunoblot (Figure 7a) while control cells did not exhibit activation of caspase 3 (Figure 7a). Activation of caspase complex inhibited the DNA repair enzyme PARP, displaying a cleaved protein band size of 89 kDa upon CAPs and topotecan treatment (Figure 7a). Thus, activation of apoptotic initiator complex caspase 9 further activated the caspase 3 executioner complex downstream and caused irreversible cell death (Figure 7b).

### 3.8. CAPs Exhibit Synergism with Topotecan

As topotecan and CAPs displayed similar mechanisms of anti-cancer activity, we sought to investigate the possible synergistic effect of the two peptides with the chemotherapeutic agent, by checkerboard assay. The percentage of viable WERI-Rb1 cells treated with sublethal concentrations of CAPs and topotecan are shown in terms of heat maps (Figure 8a,b). No antagonistic effect (i.e., higher % growth than individual drugs) was observed at ½× IC_50_ and ¼× IC_50_ values of combination drugs, indicating synergistic interactions between peptides and topotecan. A synergistic effect was also evident when plotting results in terms of the fraction of IC_50_ values (Figure 8c). Determination of FICi values for both HC3 and HC5 in combination with topotecan yielded a value of 0.5, indicating a synergistic interaction between peptides and topotecan.

## 4. Discussion

The emergence of drug resistance is a common phenomenon associated with antibacterial and anticancer drugs. In addition, the lack of selectivity of chemotherapeutic agents and the concomitant side effects warrant the development of new therapeutic strategies for combating antimicrobial/anticancer resistance. We report here the de novo design of wholly cationic peptides with antimicrobial and anticancer properties. The designed peptides displayed potent antimicrobial activity against both Gram-positive and Gram-negative strains. The peptides elicited rapid loss of bacterial membrane potential, inducing bactericidal activity. Such membranolytic activity is advantageous in averting the evolution of antimicrobial resistance owing to the increased fitness cost associated with the modification of the entire membrane by the microorganisms. 

CAPs are a promising novel class of therapeutics that were tested on a broad range of cancer types and reported to induce cancer cell death [47]. Rb is an intraocular pediatric cancer that develops in the retinal tissue at the posterior of the eye [48]. Intra-vitreal chemotherapy is a widely accepted clinical procedure for the management of advanced Rb tumors with vitreous seeds and prevents the risk of metastasis of Rb [49,50]. Topotecan is one of the chemotherapy drugs of choice for advanced cases of Rb, due to its high pharmacological activity and limited toxicity at low doses [51]. However, like many other drugs of this class, the use of high concentrations of topotecan has been reported to cause variable responses and bystander toxicity to the retina [52]. Therefore, there is an urgent clinical need to enhance the sensitivity of tumor cells to established chemotherapeutic regimens by combinatorial modalities with other drugs that function synergistically. 

In our study, among the panel of six CAPs, two peptides- HC3 and HC5 displayed a significant reduction in Rb cell proliferation. CAPs carry unnatural amino-acids with D-or-L lysine or arginine repeats on their outer surface and their net positive charge enables them for rapid accumulation around negatively charged phospholipids present on the cancer cells, forming non-covalent interactions with the side chains of lysine or arginine [53]. This mechanism of electrostatic interaction favored CAPs’ permeabilization into Rb cancer cells and inhibited their growth. HC3 and HC5 at low concentrations were able to restrain Rb cell proliferation. However, other peptides (HC1, HC2, HC4, and HC6) along the panel did not exhibit similar levels of cytotoxic effects on WERI-Rb1 cells, suggesting that the peptides’ structural configuration likely has a bearing on their function. CAPs at high doses displayed no toxicity to ARPE-19 cells, unlike chemotherapy drugs, underscoring their potential as a tumor targeting, healthy retina-sparing modality. To test the ability of CAPs to function in combination with chemotherapy, we selected the drug topotecan, which is a known topoisomerase inhibitor that causes DNA double-strand breaks [54]. Our study revealed the signaling cascade by which HC3 and HC5 selectively induced DNA damage in Rb cells. CAPs inhibited the expression of cell survival marker Bcl2 potentially disrupting the balance between cell survival and death pathways. CAPs accelerated mitochondrial dysfunction through Bax activation in Rb cells and facilitated the extrusion of cytochrome c into the cytoplasm, which further activated a regulatory network of caspases to induce cell death. The ability of damaged mitochondria to initiate apoptotic events depends on the structural and functional differences between healthy and malignant cells [55]. Downstream of mitochondrial damage, CAPs activated caspase 9 and caspase 3 to initiate the intrinsic apoptotic pathway that cleaved PARP, an enzyme essential for DNA repair and chromatin remodeling [56]. DNA repair mechanism was halted with the inhibition of PARP, and this sensitized Rb cells to CAPs. In the checkerboard assays, CAPs exhibited a synergistic interaction with the chemotherapy drug topotecan and thus CAPs can be used to enhance the activity of topotecan at a low dose. The synergism likely stems from the ability of both drugs to enhance DNA damage.

Our findings suggest that CAPs would be beneficial to enhance the chemosensitivity of low dose topotecan in combinational therapies to control Rb tumors. CAPs’ peculiar mode of action to selectively cause DNA damage, disrupting mitochondrial membrane potential and cell death in cancer cells in vitro, favors their choice as a potential adjuvant option to manage Rb or a broad range of other cancers. The advantages of using combination drugs include higher efficacy than monotherapy, reduced required dosage of chemotherapeutic drugs to achieve a positive response, and concomitant reduced bystander toxicity. However, the therapeutic efficacy of CAPs must be tested in pre-clinical models of Rb to determine its efficacy in vivo, and its absorption and diffusion into the tumor mass for further clinical consideration.

## 5. Conclusions

Our study shows that cationic antimicrobial dodecapeptides can be exploited as potential therapeutic agents in retinoblastoma as monotherapy as well as adjunctive therapy to enhance the effectiveness of currently used treatment modalities.

## Figures and Tables

**Figure 1 pharmaceutics-14-02507-f001:**
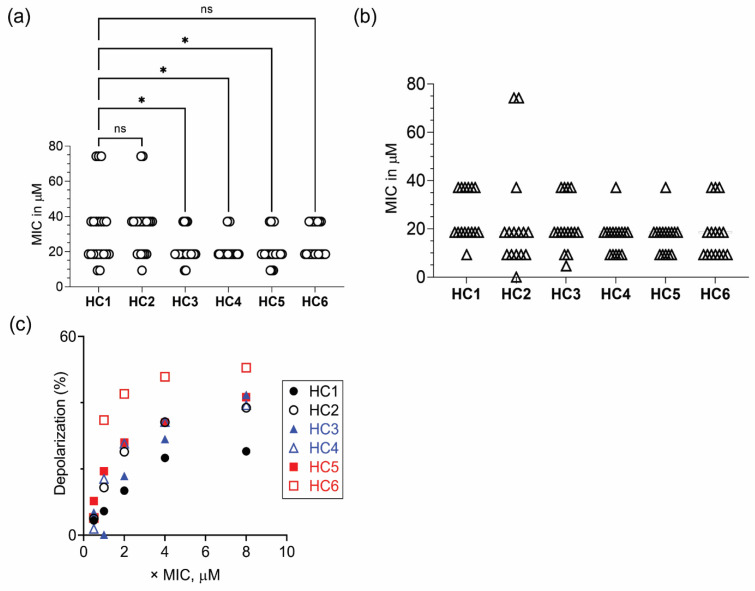
Antimicrobial properties of CAPs. Scatter plot showing the MIC values of CAPs against (**a**) *P. aeruginosa* (white circle ◯ represents *n* = 21 strains) and (**b**) *S. aureus/MRSA* (white up-pointing triangle △ represents *n* = 16 strains). (**c**) Peptide concentration-dependent increase in depolarization of *P. aeruginosa* ATCC 9027 cells. * *p* < 0.05, ns = non-significant.

**Figure 2 pharmaceutics-14-02507-f002:**
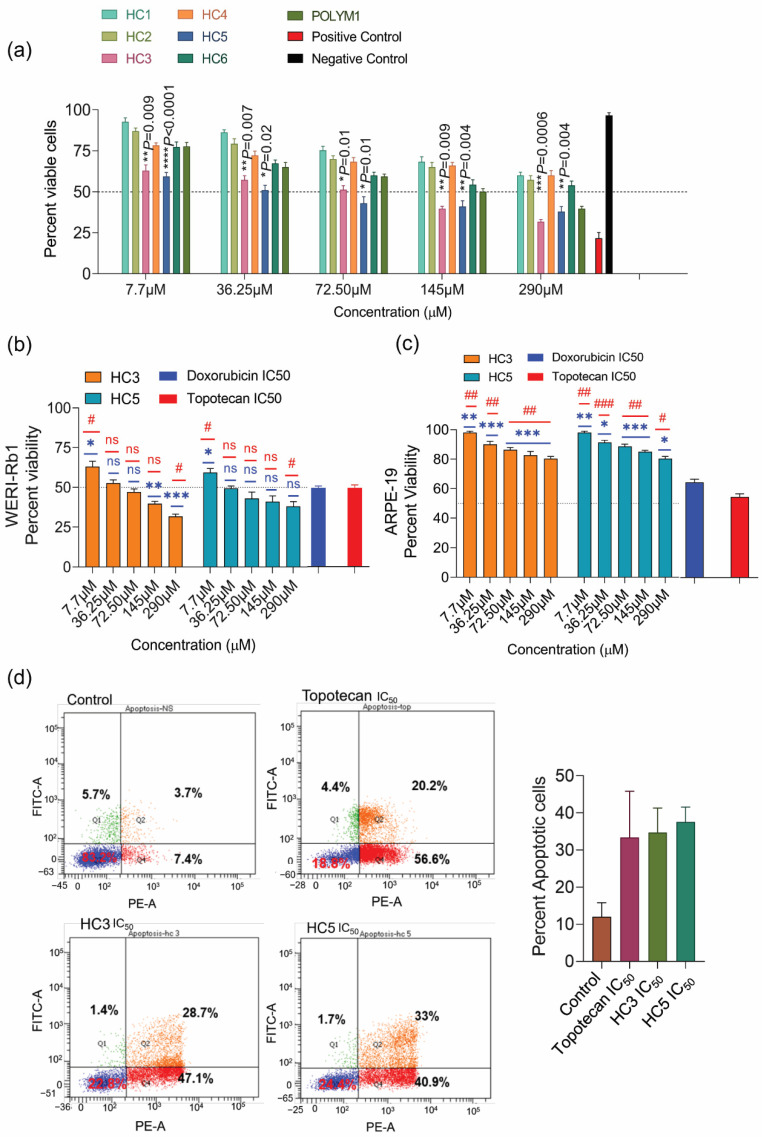
CAPs display potent anticancer activities on Rb cells. (**a**) Trypan blue cell count assay showing a reduction in WERI-Rb1 cell viability upon treatment with a different class of CAPs for 48 h. Untreated cells represent negative control, and Nocodazole (10 µM) treatment represents positive control. (**b**) MTS-based cell proliferation assay showing the response of HC3, and HC5 in WERI-Rb1 cells treated for 48 h. Doxorubicin and topotecan were used as positive controls. (**c**) MTS-based cell proliferation assay showing CAPs’ response in ARPE-19 cells treated for 48 h. Doxorubicin and topotecan were used as positive controls. (**d**) Annexin V/PI assay showing the percentage of apoptotic cells in CAPs treated cells. Values represent the mean ± s.d of three independent experiments. Two-way ANOVA with Tukey’s multiple comparisons test (for >2 groups) was used for statistical analysis. * *p* < 0.05, ** *p* < 0.01, *** *p* < 0.001, **** *p* < 0.0001, ^#^
*p* < 0.05, ^##^
*p* < 0.01, ^###^
*p* < 0.001, ns = non-significant.

**Figure 3 pharmaceutics-14-02507-f003:**
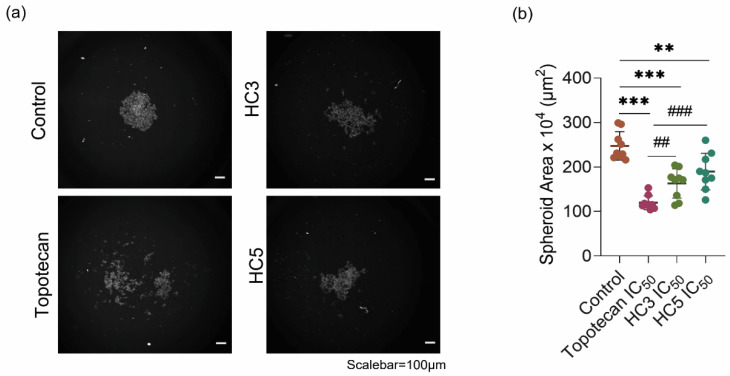
CAPs reduce Rb tumor spheroid assembly and growth. (**a**) Phenotypic spheroid assay readouts of multiple treatment outcomes as visualized by HCA microscopy. (**b**) Cumulative measurement of the spheroid area after treatments as quantified by HCA. Inhibitory concentration of peptides (HC3 IC_50_ = 77.2 µL/mL and HC5 IC_50_ = 65.2 µL/mL) and topotecan (IC_50_ = 10 nM). Values represent the mean ± s.d of three independent experiments. Two-way ANOVA with Tukey’s multiple comparisons test was used for statistical analysis. ** *p* < 0.01, *** *p* < 0.001, ^##^
*p* < 0.01, ^###^
*p* < 0.001.

**Figure 4 pharmaceutics-14-02507-f004:**
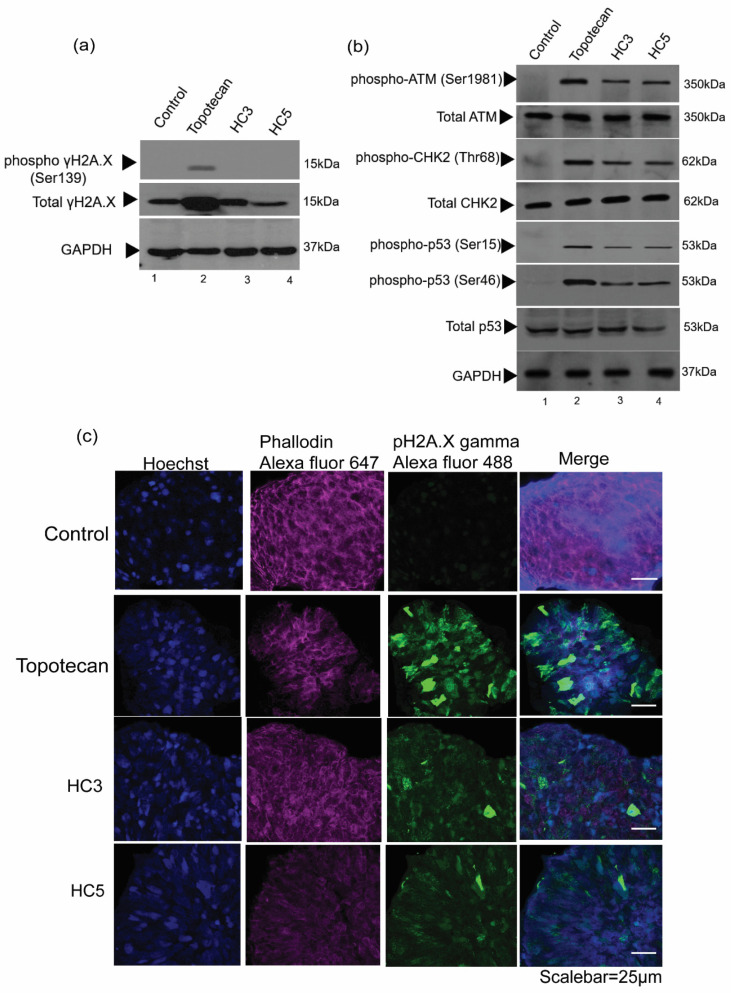
CAPs induce DNA damage and apoptosis in Rb cells. (**a**) Western blot analysis showing the phosphorylation of γH2A.X (Ser139) in WERI-Rb1 cells treated with topotecan, but not in cells treated with HC3 or HC5. (**b**) Western blot analysis showing DNA damage markers upon CAPs and topotecan treatment. GAPDH was used as loading control in Western blots. (**c**) Immunofluorescence images showing expression pγH2A.X on WERI-Rb1 spheroids treated with HC3, HC5, or topotecan. Fluorescent intensity at 488 nm (green) represents phosphorylation of γH2A.X in Rb spheroids. Phalloidin Alexa Fluor 647 represents cytoplasm (actin) marker and Hoechst 33322 stains for nuclei.

**Figure 5 pharmaceutics-14-02507-f005:**
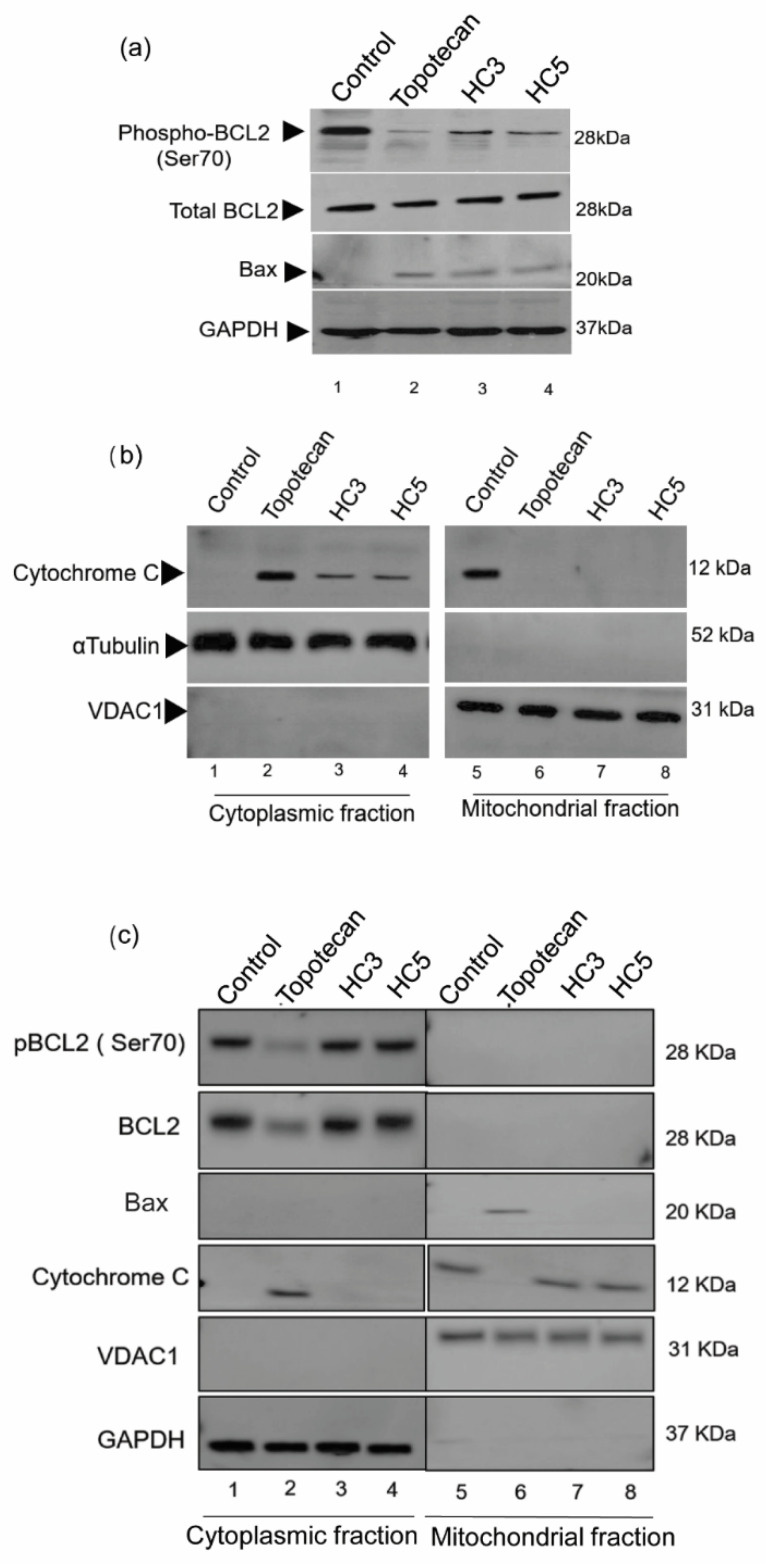
CAPs selectively disrupt the pro-apoptotic and anti-apoptotic balance in WERI-Rb1 cells. (**a**) Western blots showing the induction of mitochondrial damage in WERI-Rb1 cells, mediated by CAPs by reducing Bcl2 Ser70 phosphorylation and increasing Bax expression. GAPDH was used as a loading control. (**b**) Sub-cellular fractions of cytoplasm and mitochondria showing cytochrome c localization upon treatment with CAPs (HC3, HC5) or topotecan for 48 h. αTubulin was used to confirm cytoplasmic fraction loading control and VDAC1 as a mitochondrial fraction loading control in the blots. (**c**) Western blots showing expression of pBcl2(Ser70)/Bcl2, Bax and released cytochrome C in cytoplasmic and mitochondrial fractions of ARPE-19 cells following 48-h exposure to CAPs (HC3, HC5) or topotecan. GAPDH was used as a cytoplasmic fraction loading control and VDAC1 as a mitochondrial fraction loading control.

**Figure 6 pharmaceutics-14-02507-f006:**
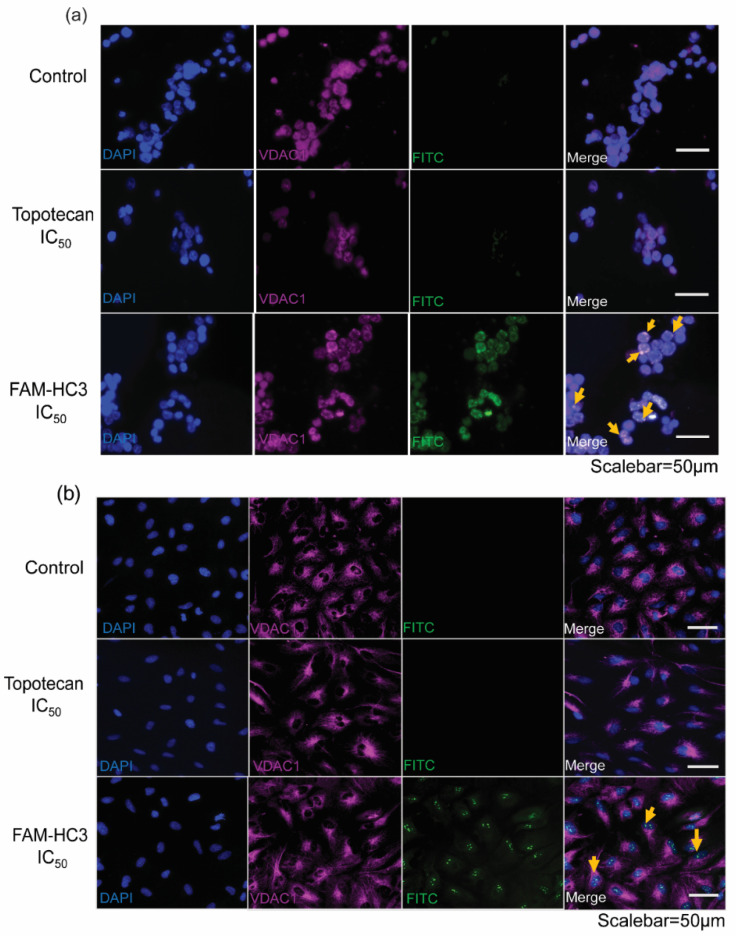
CAPs localize to mitochondria in WERI-Rb1 cells and initiate programmed cell death. (**a**) WERI-Rb1 cells, and (**b**) ARPE-19 cells were treated with topotecan IC_50_ and FAM-HC3 IC_50_ for 48 h. FAM-HC3 signal was captured at 488 nm (green). VDAC1 was used as a mitochondrial outer membrane marker and the signal was captured at 570 nm (red). The nucleus was stained with Hoechst 33322 (blue). Yellow arrows show the co-localization of FAM-HC3 with VDAC1. The scale bar is 50 µm.

**Figure 7 pharmaceutics-14-02507-f007:**
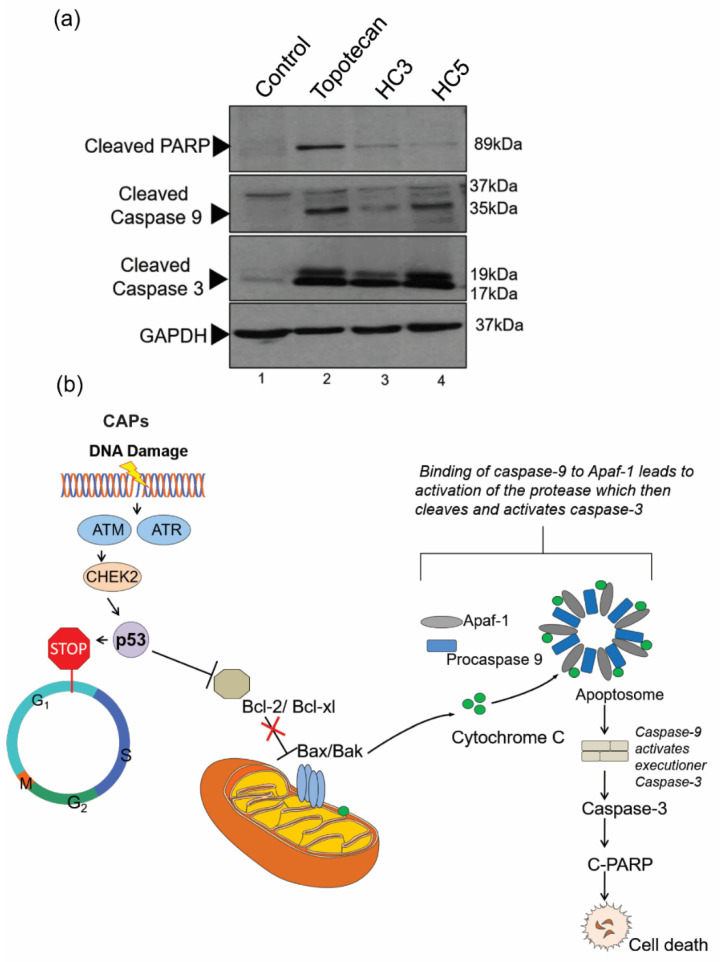
CAPs drive apoptosis through the intrinsic pathway. (**a**) Western blot analysis showing caspase-mediated apoptosis initiated by CAPs (HC3, HC5) in retinoblastoma cells. Cells were treated with topotecan (10 nM) as a control. A total of 20 µg cellular lysates were loaded into the SDS PAGE for Western immunoblot analysis. GAPDH was used as the loading control. The Western blots show activation of caspase 9, caspase 3, and PARP, indicating an intrinsic apoptotic fate in retinoblastoma cells upon peptide treatment. (**b**) Schematic illustration of intrinsic apoptosis in retinoblastoma mediated by CAPs.

**Figure 8 pharmaceutics-14-02507-f008:**
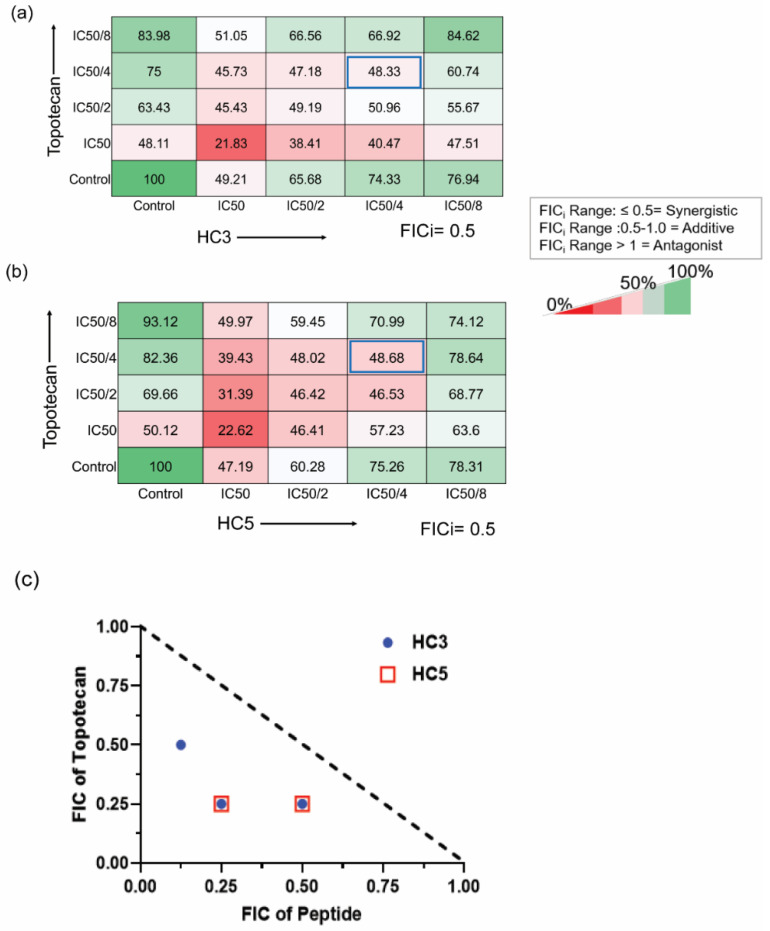
CAPs exhibit synergism with topotecan (**a**) Checker-board assay showing the drug interaction of HC3 with topotecan. FICi values were 0.5. (**b**) Checkerboard assay showing the interaction of HC5 with topotecan. (**c**) Isobologram showing the changes in the fraction of IC_50_ values for the combined drugs. The broken line represents the hypothetical additive effect. The Fractional Inhibitory Concentration Index (FICi) was determined by the combined FIC value for each peptide and topotecan (FICi = FIC of peptide + FIC of topotecan). FICi value of ≤0.5 was considered synergistic; a value of >0.5–1 indicated an additive effect of the two drugs, and a FICi value of >1 indicated antagonism. FICi values of HC3 and HC5 with topotecan are ≤0.5. Values represent the mean of percent viability.

**Table 1 pharmaceutics-14-02507-t001:** Amino-acid sequence of the designed peptides.

Peptide	Sequence
HC1	KRKRKRKRKRKR
HC2	*K*R*K*R*K*R*K*R*K*R*K*R
HC3	K*r*K*r*K*r*K*r*K*r*K*r*
HC4	KrKrKrKrKrKr
HC5	kRkRkRkRkRkR
HC6	*k*R*k*R*k*R*k*R*k*R*k*R

The amino acid residues are labelled as: *K*–ε-L-lysine; *k*–ε-D-lysine; k–D-lysine and r–D-arginine.

## Data Availability

All data and source files will be available on request.

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
