# Peer review of "Selective Induction of Intrinsic Apoptosis in Retinoblastoma Cells by Novel Cationic Antimicrobial Dodecapeptides"

_pharmaceutics, 2022, doi:10.3390/pharmaceutics14112507_

Round 1
Reviewer 1 Report
This manuscript by Chosh and coworkers reports the investigation of selective induction of intrinsic apoptosis in retinoblastoma cells by a panel of cationic antimicrobial dodecapeptides (CAPs). By means of cellular phenotype characterization, Western blotting and fluorescence imaging, the authors demonstrated that among the panel of hyper-charged cationic CAPs, HC3 and HC5 could selectively kill the significant number of retinoblastoma WERI-Rb1 cells by inducing apoptosis in WERI-Rb1 cells via activation of caspase 9 and caspase 3, cleaving the PARP protein and triggering the phosphorylation of p53 and gamma-H2A.X. Importantly, the authors revealed that HC3 and HC5 with the standard chemotherapeutic drug topotecan shows synergistic anti-cancer activities. Given that the use of high concentration of topotecan causes variable responses and toxic side-effects, this work provides evidences that the cationic CAPs deserve to be further exploited as potential therapeutic agents in retinoblastoma as monotherapy or as adjunctive therapy to enhance the effectiveness of currently used therapeutic drugs. I would recommend publication of this manuscript in Pharmaceutics subject to addressing a few concerns as follows by the authors.
1. In page 7 of 19, the data shown in Figure 1a and Figure 1b are not understandable, at least to me. The authors need to explain in the caption what the shapes and sizes of the patterns indicate. Otherwise, one cannot understand why there are statistically difference in the MICs of HC1 and HC3, HC4, or HC5, and why there are no significant changes in the MICs of HC1 and HC6 as shown on the top of Figure 1a.
2. In page 10 of 19, line 374, why “however” there?
3. In page 13 of 19, I notice that for WERI-Rb1 cells, the signals of DAPI staining nuclei, VDAC marking mitochondria and FAM-HC3 appear overlap each another. Does this mean mitochondria and nuclei of the WERI-Rb1 cells were damaged, leading to the fusion of both cellular organelles (mitochondrion and nucleus)? The authors should discuss this issue.
4. As the authors stated in page 15, the FICi is the value with which the authors judged whether the tested peptides have synergistic functions or not with topotecan. However,the authors did not state how they calculated FICi values. In my opinion, the authors should give the equation or the method for calculating or measuring FICi values.
Reviewer 2 Report
Selective Induction of Intrinsic Apoptosis in Retinoblastoma Cells by Novel Cationic Antimicrobial Dodecapeptides
Vishnu Suresh Babu1,2, Atish Kizhakeyil3, Gagan Dudeja4, Shyam S. Chaurasia5, Amudha Barathi Veluchami6, 4 Stephane Heymans2, Navin Kumar Verma3,6,*, Rajamani Lakshminarayanan6,* and Arkasubhra Ghosh1,6,*
Incorrect use of hyphen:
Line 24 “mem-brane”
Line 281 “link-age”
Line 299 Wording: “MIC value of HC3, HC4 and HC5 was statistically significant for the HC1 peptide.” The authors likely mean significant when compared to HC1 peptide
Figure 1C – Indicate if a statistical test was performed
Reviewer 3 Report
The manuscript "Selective induction of intrinsic apoptosis in Retinoblastoma cells by novel cationic antimicrobial dodecapeptides" describes a biological evaluation of six cationic oligopeptides. The introduction is well-written and the study design is interesting. I would only suggest to add a reference cationic antimicrobial peptide for comparison in the MIC determination (Table S1) and some minor editorial corrections, for example, the names of the strains should be written in italics.
To sum up, I recommend the paper for publication in Pharmaceutics after minor revisions.
